# Milpa Diet for MASLD in Mesoamerican Populations: Feasibility, Advantages, and Future Perspectives

**DOI:** 10.3390/life15050812

**Published:** 2025-05-19

**Authors:** Aline Huerta-Álvarez, Mariana Arellano, Clyo Anahí Chávez-Méndez, Paulina Carpinteyro-Espin, Carmen Palacios-Reyes, Juanita Pérez-Escobar

**Affiliations:** 1Department of Nutrition, Hospital Juárez of Mexico, Mexico City 07760, Mexico; nutriologaaline@gmail.com; 2Center of Research in Nutrition and Health, National Institute of Public Health, Cuernavaca 62100, Mexico; mc.arellano99@gmail.com; 3Private Nutrition Practice, Hospital Angeles Mocel, Mexico City 11850, Mexico; nc.clyochavez@gmail.com; 4Department of Transplantation, Hospital Juárez of Mexico, Mexico City 07760, Mexico; paucarpi@gmail.com; 5Departamento de Ciencias Médicas, División de Ciencias de la Salud, Universidad de Guanajuato, León de los Aldama 37320, Mexico; cyapalacios@gmail.com

**Keywords:** milpa diet, MASLD, antioxidants, proteins, fiber

## Abstract

Metabolic dysfunction-associated steatotic liver disease (MASLD) is the leading cause of chronic liver disease, linked closely to metabolic syndrome and rising obesity rates. Affecting up to 37% of the global adult population, MASLD prevalence is exceptionally high among individuals of Hispanic descent, with genetic factors such as the PNPLA3 gene mutation playing a significant role. The subject of this review is the traditional Mesoamerican “milpa” diet, which includes unprocessed local crops like maize, beans, pumpkins, chili, and tomatoes and may represent a strategy to combat MASLD. Current treatment recommendations emphasize weight loss; a reduced intake of saturated fats, processed meats, and added sugars; and increased physical activity. The milpa diet, rich in protein, fiber, vitamins, and bioactive compounds, aligns with these recommendations and could potentially mitigate MASLD by preventing liver fat accumulation and fibrosis. This narrative review focuses on available preclinical and clinical studies adopting the milpa diet as a culturally relevant, nutritious, and sustainable dietary approach in preventing and treating MASLD. More clinical studies are needed to develop precise nutritional quantitative recommendations and guidelines.

## 1. Introduction

Metabolic dysfunction-associated steatotic liver disease (MASLD) is the latest term for steatotic liver disease associated with metabolic syndrome, recently adopted as nomenclature. MASLD is the most common cause of chronic liver disease and is the leading cause of liver-related morbidity and mortality [1,2]. MASLD is considered a current pandemic, affecting 30% of the adult population worldwide, with an increasing prevalence, rising from 22% to 37%, from 1991 to 2019, respectively, and this is expected to continue in the following years [3,4]. This trend goes hand in hand with the increase in obesity and related diseases. Metabolic dysfunction-associated steatohepatitis (MASH) is the most severe form of MASLD, is histologically diagnosed based on the presence of lobular inflammation and the ballooning of hepatocytes, and is associated with an increased risk of fibrosis progression, which is a target in prevention and treatment [3].

When the prevalence of MASLD has been analyzed among different ethnicities, it is found to be higher among populations of Hispanic origin, including patients of Mexican origin (33%), considering genetically determined mechanisms [5]. Mutations in the patatin-like phospholipase domain containing 3 (PNPLA3) gene are higher among the Hispanic population, specifically Mexicans (49%) [6]. The risk associated with the PNAPLA3-I148M variant involves resistance to proteasomal degradation and the accumulation of lipid droplets in hepatocytes compared to the population without this polymorphism. The PNPLA3 variant also enhances the profibrogenic characteristics of hepatic stellate cells, thus not only leading to a 3.5-fold increased risk of developing hepatic steatosis but also contributing to disease progression and the development of hepatocarcinoma, which cannot be attributed solely to the progression of advanced fibrosis [7]. There is a high prevalence rate of the G risk allele in Mexicans, described as being as high as 77%. The presence of PNPLA3 polymorphisms in the Mexican population has been associated with an increased risk of MASLD, and it has been found that carriers of the GG genotype have a 3.8-fold increased risk of MASLD and a 2.3-fold increased risk of liver fibrosis, according to the pathophysiological elements previously mentioned [8].

The increasing prevalence of MASLD is closely linked to lifestyle, characterized by the widespread adoption of Western diets that include a high consumption of ultra-processed foods [4]. In this sense, the traditional Mesoamerican diet, commonly called the “Dieta de la milpa” (milpa diet), based on fresh and unprocessed local crops, could represent a viable dietary alternative to mitigate the effects of MASLD.

This review aims to summarize the nutritional characteristics of the milpa diet and the benefits that its consumption may lead to in patients with MASLD. We also analyze how this dietary pattern can benefit this population and how the bioactive components present in the milpa diet could improve their health.

## 2. Current Treatment for MASLD

Most factors contributing to the development and progression of MASLD are modifiable, so lifestyle interventions, including diet and physical activity, represent the cornerstone of its treatment. However, there is no definite pharmacological or interventional treatment.

The main goal in overweight and obese patients with MASLD is to achieve weight loss, which is suggested to be at least 5% to improve intrahepatic fat, 7 to 10% in overweight and obese patients to improve inflammation, and a goal of 10% weight loss if fibrosis is to be reversed. For lean patients, the proposed weight loss is 3 to 5% [5].

Patients are also recommended to reduce their intake of saturated fats and avoid processed red meat and added sugars in their diet, instead choosing healthy eating patterns high in fiber, such as the Mediterranean diet (MedDiet), as specified by associations such as the American Association of Clinical Endocrinology in its 2022 guidelines (AACE), European Association for the Study of the Liver (EASL) in 2021, the 2019 recommendations of the European Society for Clinical Nutrition and Metabolism (ESPEN), and the latest guidelines published in 2023 by the American Association for the Study of Liver Diseases (AASLD) [6].

In studies performed in mice, the influence of a high-fat diet on the microbiota and its relationship with metabolic risk factors have been studied, and it has been found that there is a decrease in circulating fatty acids in the ileum, altering Takeda G-protein-coupled receptor 5 (TGR5) signaling, which partly explains the altered glucose metabolism and obese phenotypes. This demonstrates the role of diet in the gut microbiota and provides a treatment target that should be included in future management strategies for patients with MASLD [7].

Trials evaluating the efficacy of probiotic use in MASLD are inconclusive because they vary in duration (4 to 56 weeks) and use different strains, mainly *Lactobacillus* and *Bifidobacterium*. In a meta-analysis *(n* = 12,682), a beneficial effect was seen on liver fibrosis, levels of alanine aminotransferase (ALT), aspartate aminotransferase (AST), and high-sensitivity C-reactive protein (hs-CRP) and tumor necrosis factor alpha (TNF-α) [8].

Regarding coffee consumption, several studies have demonstrated a protective effect in the progression of MASLD from consuming at least three cups of coffee per day. There are statistically significant protective effects of coffee consumption on the incidence of MASLD in population-based follow-ups and a reduction in progression to MASH, hepatocellular carcinoma, and cirrhosis in patients diagnosed with MASLD [9].

The avoidance of alcohol consumption is recommended to decrease the progression of MASLD and the risk of hepatocellular carcinoma, even at doses established as moderate consumption (<2 drinks per day for men and <1 for women) [5]. In individuals with MASLD, low-to-moderate alcohol consumption, defined by the authors of the article as 5 to 9 drinks and 10 to 13 drinks, respectively, increases the risk of fibrosis assessed by transient elastography and by noninvasive tests such as the FibroScan-AST (FAST) score, Fibrosis-4 (FIB-4) Index for Liver Fibrosis, and AST to Platelet Ratio Index (APRI) [10].

Physical activity is another mainstay of treatment for MASLD. It is recommended that periods of sedentary time be reduced and that an individualized exercise program of at least 150 min of moderate exercise or 75 min of vigorous exercise combined with resistance exercise per week be implemented [11]. The mechanisms by which exercise improves prognosis in this population are diverse and include improved body composition, lowering the presence of dysbiosis, a reduction in intrahepatic fat >30%, and reduced fibrosis and endothelial dysfunction [12].

Recent clinical guidelines have included direct pharmacologic treatment to address the comorbidities present in MASLD. For type 2 diabetes and obesity, glucagon-like peptide-1 receptor agonists (GLP1RAs) such as semaglutide, liraglutide, and dulaglutide can be used, as well as tirzepide, which acts as a coagonist. Other drugs that can be used are sodium-glucose cotransporter 2 inhibitors (SGLT2), such as empaglifozin or dapaglifozin, as well as metformin and insulin. For dyslipidemia, statins are recommended. In the case of obesity, bariatric surgery is also considered as an option if other interventions are not successful. In cases with MASH where resmetirom is available, its use is encouraged, while in cases that have progressed to cirrhosis or in the presence of hepatocellular carcinoma, the option of transplantation should be considered [5].

## 3. Milpa Diet Characteristics

The milpa system (MS) is a traditional agricultural production system in Mesoamerica, particularly in Mexico, that has been used since pre-Hispanic times. Its name comes from the Nahuatl language (milli, sown plot; pan, on). It is a polyculture in which maize (*Zea mays* L.) is the main crop, accompanied by beans (*Phaseolus vulgaris* L.), pumpkin (*Cucurbita pepo* L.), chili (*Capsicum annuum* L.), and tomato (*Solanum lycopersicum* L.) [13].

Considered the first organized agricultural system in the Americas, MS has been displaced by monocultures, despite offering benefits such as the more efficient use of water, land, and space and improved climate resilience [14]. It also provides greater dietary diversity and contributes to sustainable production. Its products are rich in protein, fiber, vitamins, and bioactive compounds, which have health benefits, helping to prevent diseases such as cancer, type 2 diabetes, hypertension, and obesity [15]. Based on traditional Mesoamerican crops, the milpa diet is rich in antioxidants and micronutrients due to the variety of fresh, local, and unprocessed foods that make up the diet [16]. These foods contain antioxidant compounds such as flavonoids, carotenoids, and vitamins A, C, and E, which protect cells from free radical damage and help to reduce the risk of chronic diseases [17].

Using traditional foods, Figure 1 proposes a plate layout with half of the plate filled with vegetables, a quarter with protein sources, and the rest with whole grains, cereals, and tubers. Healthy fats and beverages without sugar are included to complement the main foods. Also, it includes some examples of dishes that can be prepared daily within the milpa diet following this distribution.

Figure 2 shows the remaining food groups as snacks, and the main part of this meal contains fruits, which are complemented with seeds and nuts. Both groups use groups from the region.

Besides encouraging a diet rich in natural nutrients, the milpa diet promotes healthy habits by emphasizing the importance of an active lifestyle. In addition, it highlights that physical activity must meet specific characteristics to be considered beneficial, promoting both mental and physical health. In this sense, it is suggested that physical activity should be safe, inclusive, and playful rather than competitive, considering each person’s needs and circumstances [15].

In contrast, the Western diet is a risk factor for the development of MASLD. In Table 1 below, there is a comparison of the main characteristics of the milpa diet and the Western dietary pattern.

**Table 1 life-15-00812-t001:** Main characteristics of the milpa diet and the Western diet.

Aspect	Milpa Diet	Western Diet
Origin	The traditions of Mesoamerica are based on the millenial agricultural system	Predominant in industrialized countries, characterized by consuming processed and ultra-processed foods
Characteristic foods	Corn, beans, squash, chili, quelites, tomatoes, amaranth, ricotta cheese	Red meats, ultra-processed products, refined flours, added sugars, full-fat dairy products
Main source of proteins	Legumes (beans), insects, ricotta cheese, and lean meats	Red meats, sausages, full-fat dairy products, and animal proteins
Predominant carbohydrates	Nixtamalized corn and amaranth	Refined flours and added sugars
Fats	Predominantly unsaturated (seeds, avocado, zucchini, sunflower oil)	High in transfats and saturated fats
Fiber	High in fiber (whole grains, vegetables, pulses)	Low in fiber (diets high in processed and refined foods)
Caloric density	Moderate, based on natural and minimally processed ingredients	High, with rich calorie load from fats and sugars
Health impact	Promotes cardiovascular health, prevents metabolic diseases by reducing adipose tissue at the central level	Linked to obesity, type 2 diabetes, hypertension, and cardiovascular disease
Sustainability	Environmentally friendly, based on local production and polycultures	High environmental impact due to meat consumption and intensive monoculture
Food processing	Minimally processed, fresh, and natural food	High, with additives, preservatives, and excess sodium

Adapted from [15,18,19,20,21].

## 4. Potential Benefits of Milpa Diet Components

The components of the milpa diet, such as fiber, plant-based proteins, antioxidants, vitamins, and minerals, among others, could offer significant benefits for patients with MASLD due to their content of bioactive molecules that improve and optimize the performance of adipose tissue and the liver, representing a suitable prophylactic and therapeutic alternative [22].

The following sections highlight the benefits in MASLD gained from the types of macronutrients, fiber, specific nutrients, and foods proposed in the milpa diet. Further studies are needed to support the efficacy of the milpa diet for the treatment of MASLD. The main findings of preclinical and clinical studies focused on the bioactive compounds in the milpa diet are summarized in Appendix A.

### 4.1. Suggested Protein Sources and Their Benefits

The consumption of plant-based protein, mainly from legumes, nuts, and seeds, plays a role in the nutritional management and prevention of MASLD, due to its advantages in controlling body weight, decreasing intrahepatic fat content, and reducing nutrient intake linked to the development and progression of hepatic steatosis [23]. Although plant-based diets have several metabolic benefits, not all have the characteristics compatible with a healthy diet. Therefore, indexes have been created for quality evaluation, increasing their quality score according to their richness in whole grains, fruits, vegetables, legumes and seeds. Diets based on plants with a higher nutritional quality index are protective against hepatic steatosis, obesity, central adiposity, and HOMA index [24]. Diets dominated by refined grains and added sugars in foods and beverages have been linked to increased intrahepatic fat accumulation [25].

Legumes were previously considered ‘incomplete protein sources’; however, it has now been shown that there is complementarity between the various sources of essential amino acids, such as legumes and cereals, even if they are not eaten at the same meal time, e.g., beans and nixtamalized maize. In addition, preparation techniques such as soaking and cooking facilitate the absorption of amino acids obtained from legumes [26]. Regarding beans, a study by Bahrami et al. [27] concluded that a higher bean consumption was associated with a 35% lower risk of MASLD, even after considering other factors such as overall diet, exercise, and health conditions like diabetes and dyslipidemia.

The milpa diet also suggests the consumption of seeds such as pumpkin seed, chia, and peanut as sources of vegetable protein, which also provide antioxidants, monounsaturated fats, and fiber, key nutrients to improve the cardiovascular factors present in MASLD. The consumption of 15–30 g of seeds and nuts per day has been associated with a lower prevalence of MASLD. When comparing women who did not consume nuts with those who did, women who consumed this food group had statistically significant decreases in Fatty Liver Index score, BMI, and waist circumference and reduced glycated hemoglobin (HbA1c), insulin, and HOMA-IR [28].

In turn, another study conducted in murine models by Hussain et al. demonstrated that pumpkin seeds showed antihyperglycemic and antihyperlipidemic effects in albino rats [29]. A significant decrease was observed in blood glucose levels (128.33 mg/dL), total plasma cholesterol (88.43 mg/dL), triglycerides (69.79 mg/dL), and low-density lipoprotein cholesterol (21.45 mg/dL) in the groups of rats fed 15 g of pumpkin seed powder.

One of the dairy products recommended for consumption in moderate amounts in the milpa diet is requeson [15]. There has been debate as to whether dairy consumption may increase the risk of MASLD due to saturated fat intake; however, a meta-analysis investigating the relationship between dairy consumption and MASLD risk found a protective effect with the consumption of dairy in general, milk, and yogurt; however, no such relationship was observed between cheese intake and MASLD. Although the mechanism by which it exerts this beneficial effect on MASLD is unknown, it has been proposed that one of the attributable mechanisms is its protein content, which could promote insulin regulation and appetite [30].

It is suggested that red meats including pork, beef, and goat meat are consumed in small quantities and at a low frequency (two times a month), avoiding the fatty parts and choosing preparations such as roasted, steamed, or baked meat, with plenty of vegetables, as in the case of dishes such as pozole or tlalpeño soup [15]. For this reason, the recommendation of the milpa diet is consistent with the guidelines on limiting red meat intake and avoiding processed red meat consumption in the context of MASLD.

A dose-dependent relationship has been demonstrated between the percentage of intrahepatic fat and red meat consumption [31]. Additionally, every 25 g of processed red meat increases the risk of MASLD by 11.1% [32]. In patients with the rs738409 polymorphism in the PNPLA3 gene combined with a high red meat intake, fibrosis (>F2) increases, which is not observed in patients with this polymorphism and a high white meat intake [33].

In a prospective study, with a follow-up longer than 10 years, it was found that replacing 80 g of red meat and sausages with legumes decreased the risk of MASLD (HR: 0.96, 95% CI: 0.94–0.98), with similar results obtained after replacing poultry with the same amount of legumes (HR: 0.97, 95% CI: 0.95–0.99) [34].

The region’s cuisine also allows for the incorporation of protein from insects, such as grasshoppers, and meats, including rabbit, as well as fish rich in omega-3 fatty acids, the benefits of which in the management of MASLD will be explained subsequently. Insects possess a protein content of up to 60% of their body weight and provide all essential amino acids and antioxidant compounds. In vivo studies have demonstrated a hypoglycemic and hypolipidemic effect, suggesting a potential alternative for regulating the various mechanisms contributing to MASLD. Notably, the production of insects is more sustainable than other protein sources, especially when compared to the process involved in red meat. However, the uptake of insect-based protein remains limited [35,36].

### 4.2. Benefits of Carbohydrates and Lipids in the Milpa Diet

The liver has remarkable metabolic plasticity, adaptively switching between energy storage and supply to maintain glucose homeostasis and ensure the proper functioning of the organism. This process depends on the action of glucoregulatory hormones, such as insulin and glucagon, which respond according to the availability and/or absence of nutritional substrates [37,38].

When there is impaired hepatic regulation of glucose metabolism, secondary to sustained inadequate and excessive supply, which may be facilitated by the frequent consumption of ultra-processed products high in saturated fats, sodium, and refined sugars, as well as genetically susceptible individuals, a pro-inflammatory environment is promoted, leading to relative pancreatic beta-cell dysfunction, triggering insulin resistance and excess adiposity, which are central to the pathogenesis of steatotic liver disease [37,38].

Therefore, the most studied and effective dietary patterns for the treatment of MASLD are those that prioritize and ensure an adequate intake of fiber and unsaturated fatty acids, with sources such as vegetables, fruits, legumes, whole grains, and cereals, as well as seeds and oilseeds [39].

Dysbiosis in MASLD is characterized by an increased Firmicutes/Bacteroidetes ratio, leading to the activation of inflammatory mechanisms, bacterial translocation, insulin resistance, and, as a consequence, increased hepatic lipid deposition and fibrosis. High-fiber dietary patterns such as the milpa diet are considered to mitigate dysbiosis by optimizing mitochondrial function, decreasing metabolic stress and inflammation. In comparison, the mechanism by which other patterns, such as the MedDiet, act is by improving insulin resistance, lipogenesis, and inflammation [40].

There are reports of dietary interventions with the characteristics described above showing beneficial effects in ectopic fat accumulation and anthropometric, biochemical, and clinical markers. These benefits may be underpinned by the nutritional framework and components characterizing these foods, including their high fiber content and low glycemic index. These food characteristics are widely recognized to address underlying insulin resistance and promote the regulation of ectopic fat storage [41].

In Mexico and Mesoamerica, maize is the primary source of carbohydrates and is one of the four main foods promoted by the milpa diet. The benefits of maize consumption in liver health, particularly in the context of the MASLD, can be attributed to its components, such as insoluble fiber content, peptides, anthocyanins. and polyphenols [42,43].

Whole grains and cereals are rich in insoluble fiber, which has been found to reduce lipid levels and inhibit hepatic fat accumulation [43], which are beneficial effects in the management of dyslipidemia and steatosis. It has also been reported that insoluble fiber promotes hepatic mitochondrial fatty acid oxidation; one of the enzymes involved in this is carnitine palmitoyl transferase-1 (CPT-1), crucial in the transport of fatty acids into the mitochondria for oxidation [42].

In animal model studies, maize peptides have been reported to help decrease the fibrosis present in MASLD in mice by inhibiting the NLRP3 inflammasome and modulating the gut microbiota, resulting in a reduction in lipid accumulation and oxidative stress, which are key factors in the pathogenesis of the disease. In addition, protective effects for liver injury have been demonstrated in humans by improving lipid profiles and reducing oxidative stress markers [44,45].

Anthocyanins contained in purple corn have shown to ameliorate chronic liver injury in experimental models by modulating oxidative stress and apoptosis pathways, including a reduction in liver enzymes and improvement in liver histology [46]. It has also been reported that polyphenol-rich extracts from maize can reduce cholesterol and triglyceride levels and prevent hepatic lipid accumulation in mice fed a high-fat diet, suggesting a potential role in managing obesity-related liver conditions [47].

Another key food in the milpa diet and a significant source of carbohydrates is amaranth, a pseudocereal that has shown multiple benefits for liver health. These favorable effects are mainly attributed to its influence on lipid metabolism, its ability to modulate the gut microbiota, its amino acid profile, and its antioxidant properties. Studies have reported that amaranth supplementation significantly reduces levels of triglycerides, total cholesterol, and phospholipids in the liver of mice fed with a high-fat diet. In addition, in ethanol-treated rats, amaranth decreased free and esterified cholesterol in the liver, probably through the increased expression of the low-density lipoprotein cholesterol (LDL) receptor and the reduced expression of HMG-CoA reductase, a key enzyme in cholesterol synthesis [48,49]. Amaranth has also shown hepatoprotective activity in experimental models, mainly due to its seeds and leaves, which contain high levels of phenolic compounds that contribute to its antioxidant capacity, helping to neutralize free radicals and reduce oxidative stress [50,51].

Among the modulating properties of the gut microbiota in animal studies, amaranth supplementation has shown to reverse the reduction in bacterial diversity and richness induced by high-fat diets, suggesting a role in improving gut health and thus liver health [49].

Other foods that are sources of carbohydrates and fiber that can be identified in the milpa diet are shown in Table 2.

**Table 2 life-15-00812-t002:** Sources of carbohydrates and fiber in the milpa diet.

Food Group	Components
Whole grains and tubers	Corn, amaranth, oats, sweet potato, cassava, chayotextle, or chinchayote.
Vegetables	Nopales, quelites, quintoniles, purslane, green beans, romeritos, huauzontle, tomatoes, citlali tomatoes, tomatillo, miltomate, chili peppers, bell peppers, squash, chayote, chilacayote, colorines, izote flower, jicama, watercress, chaya, huitlacoche, achiote, epazote, vanilla, acuyo, mushrooms, and allspice, among others.
Legumes, seeds, and oilseeds	Beans, fava beans, chia seeds, chocolate, peanuts, and pumpkin seeds.
Fruits	Soursop, prickly pear, papaya, black zapote, chicozapote, mamey, guava, tejocote, capulín, pineapple, anona, xoconostle, cherimoya, nance, berries, yellow plum, and pitahaya.

Taken and adapted from [52].

In addition to carbohydrates and dietary fiber, lipid intake should be considered quantitatively and qualitatively. A substantial body of evidence shows that saturated and trans fats have a negative impact on overall health, particularly affecting liver function. In contrast, diets rich in monounsaturated and polyunsaturated fats, especially omega-3 fatty acids, appear to offer therapeutic benefits and protective effects. These healthier fats are linked to improved liver enzyme profiles, as well as reduced inflammation and fibrosis, which together help maintain optimal liver health and prevent disease progression [53].

The primary sources of plant fats in the milpa diet include avocado, a source of MUFA, and the seeds and oilseeds described above. Avocado is a plant food source of monounsaturated fatty acids and antioxidants that may improve the lipid profile and optimize mitochondrial function and thereby mitigate the metabolic disorders associated with the disease. Avocado oil has shown beneficial effects in preclinical studies. Animal studies have shown that avocado oil supplementation improves mitochondrial function, reduces oxidative stress, and decreases inflammation, key factors in the progression of MASLD [54,55]. A secondary analysis of a randomized controlled trial in Latino adults found no significant differences in liver function biomarkers, such as GGT, hs-CRP, and the MASLD fibrosis score, between the low- and high-avocado-consumption groups. This suggests that more research is needed to confirm the clinical effects in humans [56].

The potential of w3 PUFAs, particularly eicosapentaenoic acid (EPA) and docosahexaenoic acid (DHA), in liver health has been studied. Several mechanisms have been proposed by which w3-PUFAs exert beneficial health effects. They reduce de novo lipogenesis and promote fatty acid partitioning towards β-oxidation rather than triacylglycerol synthesis, resulting in decreased hepatic triglyceride accumulation and improved hepatic steatosis levels, as demonstrated in both clinical and preclinical studies [52,57]. Additionally, research suggests that omega-3 PUFA supplementation can significantly reduce liver fat and improve key liver enzyme levels, such as ALT, AST, and gamma-glutamyl transferase (GGT) [58]. Omega-3 foods in the milpa diet include pumpkin seeds, walnuts, and chia seeds, which are excellent sources of ALA (alpha-linolenic acid) [52].

*Salvia hispanica* (chia) seeds, which are recommended in the milpa diet, contain bioactive compounds, such as vitamin E, tocopherols, carotenoids, and polyphenols, which have antioxidant properties that reduce cell damage. In addition, chia seeds have been shown to have several therapeutic benefits, such as regulating blood lipid levels, reducing systemic inflammation, and preventing cardiovascular disease. They are also one of the most essential plant sources of omega-3 fatty acids, an important anti-inflammatory agent [59].

### 4.3. Antioxidants and Micronutrients

The basis of this dietary model comprises vegetables and fruits, which provide a variety of nutrients, including vitamins and antioxidants. It is recommended that a variety of vegetables be included in each meal, in abundance and with different cooking methods, such as being served raw, blanched, boiled, or steamed, to preserve micronutrients [60]. The estimated fruit and vegetable consumption in the Mexican population is 47% below recommendations [61]. There is an inverse relationship between the risk of MALSD and a higher consumption of vegetables and whole grains as assessed by scales such as the Dietary Approaches to Stop Hypertension Score, Alternate Healthy Index Score, or Alternate Mediterranean Diet Score [62].

One antioxidant that stands out is lycopene, a fat-soluble carotenoid present in tomatoes, which are a crucial ingredient in the milpa diet and whose absorption improves when consumed in conjunction with fats, as is the case in many dishes in the region’s gastronomy. It has an antioxidant, immunomodulatory, cardioprotective, anti-inflammatory role and attenuates liver damage by reducing hepatic steatosis in individuals with MASLD [63].

Another vegetable that is a source of various bioactive compounds is nopal, which is widely used in the milpa diet. A study conducted by Morán-Ramos et al. [64] evaluated the impact of nopal consumption in obese Zucker rats (fa/fa) fed either a standard diet or a diet supplemented with 4% nopal for seven weeks. The results showed that nopal reduced hepatic triglycerides by 50%, decreased hepatomegaly, and improved biomarkers of liver damage, such as alanine and aspartate aminotransferase. Additionally, its consumption was associated with increased serum adiponectin levels and the expression of genes related to lipid oxidation and export. It also reduced hepatic oxidative stress and lipid peroxidation, improved insulin signaling, and decreased postprandial insulin concentration. This study suggests that nopal may attenuate hepatic steatosis by promoting fatty acid oxidation, very-low-density lipoprotein cholesterol (VLDL) synthesis, and insulin response.

Concerning fruits, the milpa diet suggests consuming them whole, avoiding juices or those containing added sugars, in the same way as is recommended in the treatment of MASLD, as it has been observed that the intake of natural juices or dried fruits is associated with a higher intrahepatic fat content [65].

Spices such as chili, epazote, achiote, oregano, coriander, pepper, and vanilla provide additional nutrients. Capsaicin, present in chilies, offers anti-inflammatory and antioxidant benefits and decreases lipid deposition in the liver at murine levels [66]. Additionally, it plays a role in weight regulation, insulin resistance improvement, fibrosis decrease, and progression to hepatocellular carcinoma [67].

Cacao (*Theobroma cacao*) is known for its high nutritional value, providing antioxidants, fiber, and iron. It is recommended that it be consumed without added sugar to enhance its benefits, such as lipid peroxidation, insulin resistance, and inflammation, in MASLD [68]. In vitro studies have demonstrated the signaling involved in human hepatocytes by modulating the activating fibroblast growth factor 21 (FGF21), promoting glucose homeostasis, and improving mitochondrial function and inhibiting oxidative stress and lipogenesis [69].

Cinnamon (*Cinnamomum* sp.) is one of the culinary ingredients present in Mexican cuisine, even though it is not native to the Mesoamerican region. Studies have been carried out in various populations for varying lengths of time, using doses ranging from 1.5 to 6 g per day in capsules or infusions. Although there is no standardization of the effects of cinnamon, several studies have demonstrated its hypoglycemic effects, insulin reduction, and improvements in the lipid profile during the time that cinnamon is consumed. These benefits have also been corroborated in patients with MASLD [70,71].

A recent meta-analysis by Ranasinghe et al. and a systematic review by Bandara et al. on the effects of cinnamon extracts in diabetes have demonstrated multiple benefits both in vitro and in vivo. In vitro, cinnamon has shown the potential to reduce postprandial glucose absorption by inhibiting enzymes involved in carbohydrate metabolism, such as pancreatic α-amylase and α-glucosidase. Additionally, it stimulates cellular glucose uptake through the translocation of GLUT-4 to the membrane, promotes glucose metabolism and glycogen synthesis, inhibits gluconeogenesis by affecting key regulatory enzymes, and stimulates insulin release, enhancing insulin receptor activity. The active compound responsible for these effects is cinnamylamine B1 [72,73].

The beneficial effects of cinnamon in vivo include the attenuation of weight loss associated with diabetes, reduced fasting glucose, decreased LDL cholesterol, increased HDL cholesterol, HbA1c, and increased circulating insulin levels [74]. Another study by Eidi et al. demonstrated that cinnamon administration for 28 days significantly reduced serum markers of liver damage (AST, ALT, and ALP) and increased levels of superoxide dismutase and catalase [75].

Despite the benefits of cinnamon, there have been some concerns about its safety due to its coumarin content, a compound with known hepatotoxic effects, but the content in Ceylon cinnamon, the most commonly used type of cinnamon in Mexico, is below the tolerable daily intake established by the European Food Safety Authority [76].

There are other components of the milpa diet, such as coriander, which could contribute to MASLD due to its effect on reducing weight, blood pressure, and lipids. Meanwhile, cumin provides benefits in controlling lipids, total cholesterol, glucose, and insulin [71].

Garlic contains allicin or diallyl thiosulfinate, which modulates the microbiota, reduces the production of lipopolysaccharides, and decreases triglyceride deposition in the liver [77]. A decrease in total cholesterol, LDL cholesterol, and triglyceride levels has also been observed [71]. Onion has effects on TNF-∝ levels, liver enzymes, insulin, glucose, and triglycerides, showing benefits in steatosis and liver inflammation [77]. Combining garlic and onion could enhance the benefits in patients with MASLD [78].

## 5. Discussion

As has been described above, the bioactive components of the milpa diet, such as fiber, vegetable protein, antioxidants, vitamins, and minerals, among others, may provide significant benefits to patients with MASLD and comorbidities.

A study carried out by Domínguez-Uscanga et al. [79] reported the beneficial effect of a baked snack containing 70% corn and 30% beans in reducing serum lipids by inhibiting PPAR-γ and SREBP2 in a murine model with a high-fat diet. The study suggests a reduction in obesity, dyslipidemia, and MASLD.

However, there is little information on this topic, and most of the data found are related to individual components and not to the overall dietary pattern, as is the case with the MedDiet, which has solid evidence behind it.

Studies have shown that adherence to the MedDiet is associated with a lower prevalence and severity of hepatic steatosis, as well as significant reductions in weight, body mass index (BMI), blood pressure, and intrahepatic fat content in adult patients, including benefits from the consumption of dairy products, nuts, and dried fruits (12). For example, a study within the PREDIMED trial found that a MedDiet rich in extra virgin olive oil was associated with a lower prevalence of hepatic steatosis in older people at high cardiovascular risk [80]. Another study highlighted that greater adherence to the MedDiet was significantly associated with lower rates of MASLD, as well as with a reduction in triglyceride levels and improved cardiovascular risk profiles [81].

There is only one study conducted on the Mexican population with MASLD that compared the MedDiet with a traditional Mexican diet. In both groups, significant differences were observed at the 24-month follow-up in terms of anthropometric parameters, such as weight, BMI, waist circumference, and hip circumference, as well as biochemical markers, specifically, AST and ALT [82].

Another limitation of this review is that the availability of the foods described vary according to the region of Mesoamerica; most of the foods presented are mostly available in central regions of America, such as Mexico, and a more extensive review would be needed to include native foods from other countries such as Guatemala, Belize, El Salvador, Honduras, and Nicaragua.

Therefore, it can be argued that the application of the MedDiet in the Mesoamerican population, despite presenting numerous benefits due to its traditional and nutritious ingredients, has several limitations, related mainly to food access and cultural, economic, and geographical differences; for example, the use of olive oil is an essential component of the MedDiet, but in many areas of Mesoamerica, its availability tends to be limited and it is unaffordable, since it is not a predominant local product in the region. Another food area in the diet is fish; while some coastal regions of Mesoamerica have access to fish, in inland areas or rural areas distant from the coast, it may not be readily available, and the cost may be high. In addition, some of the typical ingredients of the MedDiet, such as certain fruits (red fruits, grapes, and figs) and vegetables (eggplants and artichokes) may not be easily accessible in all regions, complicating the adoption of, and therefore the adherence to, this pattern.

Other differences identified as points in favor of the milpa diet include the fact that the MedDiet includes certain foods that are very different from Mesoamerican culinary traditions; instead of using corn, beans, chili peppers, squash, and other native ingredients, which also have a better infrastructure for distribution and storage, the MedDiet favors the consumption of bread, pasta, and wheat products, which displaces traditional preparations.

## 6. Conclusions

The increasing prevalence of MASLD is closely linked to the adoption of Western diets high in ultra-processed foods. The bioactive components of the milpa diet can significantly contribute to improving liver health and preventing the progression of MASLD, while providing a more appropriate dietary pattern compared to the MedDiet in Mesoamerican populations. Its adoption is not only culturally relevant and nutritionally balanced but also sustainable, offering health benefits such as an improvement in steatosis, reduction in fat percentage, glycemic control, and weight loss. The combination of its nutrients could provide a protective effect against the incidence and progression of MASLD due to its bioactive components and accessibility; however, more studies in clinical settings are needed to more precisely elaborate nutritional recommendations and guidelines.

## 7. Futures Perspectives

Currently, no randomized clinical trial or original study conclusively demonstrates the efficacy of the milpa diet in managing MASLD. However, by analyzing the components of this traditional Mesoamerican dietary pattern, it can be inferred that its bioactive compounds, such as fiber, antioxidants, and vitamin and mineral contents, could offer benefits for patients with MASLD.

The healthy and sustainable dietary guidelines for the Mexican population [69] align with the milpa diet recommendations, reinforcing the idea of its applicability in this context. Consequently, the inclusion of the milpa diet in the nutritional management of patients with MASLD should be considered, alongside the recommendation to conduct original studies to scientifically evaluate its efficacy in this patient population.

## Figures and Tables

**Figure 1 life-15-00812-f001:**
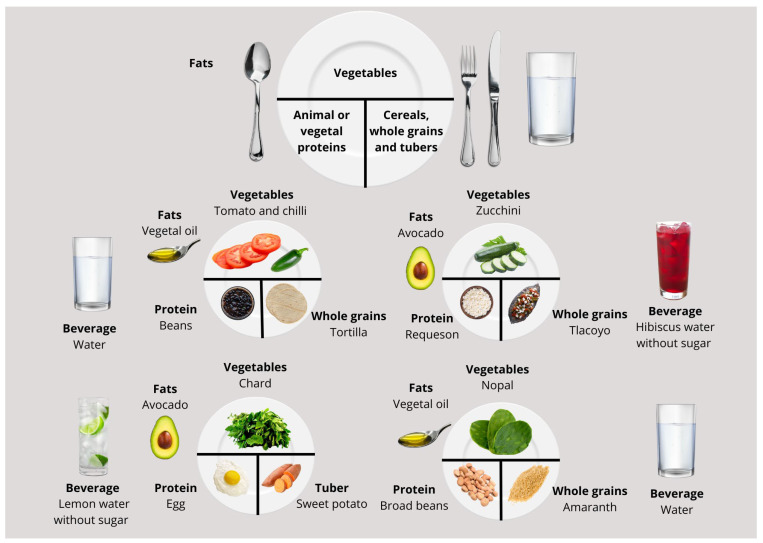
Examples of main meals using traditional foods of the milpa diet.

**Figure 2 life-15-00812-f002:**
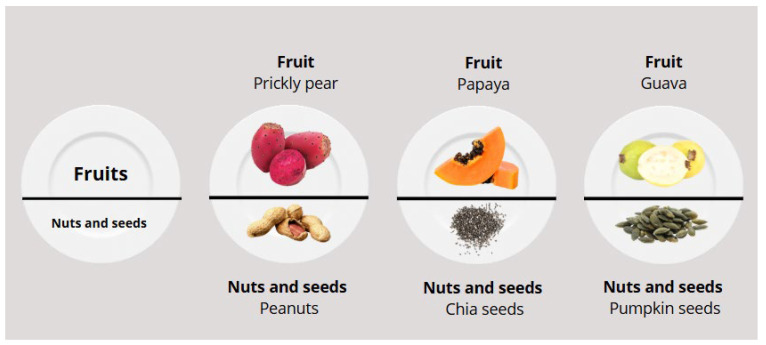
Examples of combinations of snacks for a balanced diet using traditional foods of the milpa diet.

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
