# Peer review of "Milpa Diet for MASLD in Mesoamerican Populations: Feasibility, Advantages, and Future Perspectives"

_life, 2025, doi:10.3390/life15050812_

Round 1
Reviewer 1 Report
Comments and Suggestions for Authors
The authors provide an interesting review of the Milpa diet as a potential treatment and prevention strategy for MASLD. To date, the literature has been dominated by reports of the benefits of the Mediterranean diet. However, it should be noted that the availability of individual products in different regions of the world may vary and it is worth looking for solutions on how to use local resources in a sustainable way, adapting the diet to available regional products, rather than rigidly sticking to one dietary pattern, which is not always easy to implement.
Below are some comments/suggestions:
- Chapter 2. It is worth mentioning the research on different dietary models in the treatment of MASLD and the role of the gut microbiota. https://pmc.ncbi.nlm.nih.gov/articles/PMC11722922/
Hamamah S, Iatcu OC, Covasa M. Dietary Influences on Gut Microbiota and Their Role in Metabolic Dysfunction-Associated Steatotic Liver Disease (MASLD). Nutrients. 2024 Dec 31;17(1):143. doi: 10.3390/nu17010143. PMID: 39796579; PMCID: PMC11722922.
- Line 214 - I would avoid the word "fantastic".
- Line 323 – "Cocoa (Theobroma cacao) for its high nutritional value may improve the risk for cardiovascular disease and diabetes"; are you sure you meant that this risk is increased?
- Line 331 - what type of cinnamon is used in Mesoamerican cuisine? This is interesting given the coumarin content and the research on its adverse health effects. Can you add any references here, expand on the topic?
- Abbreviations/expansions:
Acronyms/Abbreviations/Initialisms should be defined the first time they appear in each of three sections: the abstract; the main text; the first figure or table. When defined for the first time, the acronym/abbreviation/initialism should be added in parentheses after the written-out form.
Line 174 – to be explained: HbA1C (you made it in line 376), HOMA-IR
Line 263 – to be explained: LDL
Line 343 – to be explained: TNF-∝
Line 353 – abbreviation NAFLD was introduced earlier
Line 358 – LDL appears earlier in the text
Line 390 – to be explained: VLDL
- References should be written in accordance with the journal's guidelines.
Thank you and I wish you continued success in your scientific work.
Author Response
Comments 1:
The authors provide an interesting review of the Milpa diet as a potential treatment and prevention strategy for MASLD. To date, the literature has been dominated by reports of the benefits of the Mediterranean diet. However, it should be noted that the availability of individual products in different regions of the world may vary and it is worth looking for solutions on how to use local resources in a sustainable way, adapting the diet to available regional products, rather than rigidly sticking to one dietary pattern, which is not always easy to implement.
Below are some comments/suggestions:
- Chapter 2. It is worth mentioning the research on different dietary models in the treatment of MASLD and the role of the gut microbiota. lhttps://pmc.ncbi.nlm.nih.gov/articles/PMC11722922/
Hamamah S, Iatcu OC, Covasa M. Dietary Influences on Gut Microbiota and Their Role in Metabolic Dysfunction-Associated Steatotic Liver Disease (MASLD). Nutrients. 2024 Dec 31;17(1):143. doi: 10.3390/nu17010143. PMID: 39796579; PMCID: PMC11722922.
Response 1: It has been added to our job, thank you for the information.
Comments 2:
Line 214 - I would avoid the word "fantastic".
Response 2:
Thanks for your comment, we have changed it for the word "remarkable"
Comments 3:
Line 323 – "Cocoa (Theobroma cacao) for its high nutritional value may improve the risk for cardiovascular disease and diabetes"; are you sure you meant that this risk is increased?
Response 3: We edited it as we focused more on MASLD rather than the effects for other comorbidities.
Comments 4: Line 331 - what type of cinnamon is used in Mesoamerican cuisine? This is interesting given the coumarin content and the research on its adverse health effects. Can you add any references here, expand on the topic?
Response 4: This is an interesting comment. We have added this to the text.
Comments 5:
- Abbreviations/expansions:
Acronyms/Abbreviations/Initialisms should be defined the first time they appear in each of three sections: the abstract; the main text; the first figure or table. When defined for the first time, the acronym/abbreviation/initialism should be added in parentheses after the written-out form.
Line 174 – to be explained: HbA1C (you made it in line 376), HOMA-IR
Line 263 – to be explained: LDL
Line 343 – to be explained: TNF-∝
Line 353 – abbreviation NAFLD was introduced earlier
Line 358 – LDL appears earlier in the text
Line 390 – to be explained: VLDL
- References should be written in accordance with the journal's guidelines.
Thank you and I wish you continued success in your scientific work.
Response 5: We added the definition of each one of these and we avoided NAFLD as we are using the term MASLD. we made the suggested corrections, thank you
Reviewer 2 Report
Comments and Suggestions for Authors
This review focuses on the traditional Mesoamerican "milpa" diet, which includes unprocessed local crops like maize, beans, pumpkins, chili, and tomatoes, and may offer a strategy to combat MASLD. The review concludes that adopting the milpa diet could be a culturally relevant, nutritious, and sustainable dietary approach to preventing and treating MASLD, promoting better liver health and reducing the risk of associated diseases.
However, in this paper, the information is fragmented and not well organized to support the ideas or conclusions. All the aspects related to MASLD and milpa diet are talked, but in a superficial way, so the review is more like a paper for science popularization rather than academic review.
Author Response
Comments 1:
This review focuses on the traditional Mesoamerican "milpa" diet, which includes unprocessed local crops like maize, beans, pumpkins, chili, and tomatoes, and may offer a strategy to combat MASLD. The review concludes that adopting the milpa diet could be a culturally relevant, nutritious, and sustainable dietary approach to preventing and treating MASLD, promoting better liver health and reducing the risk of associated diseases.
However, in this paper, the information is fragmented and not well organized to support the ideas or conclusions. All the aspects related to MASLD and milpa diet are talked, but in a superficial way, so the review is more like a paper for science popularization rather than academic review.
Response 1:
Thank you for your feedback on the article. I appreciate the points raised. However, as you correctly pointed out, there is still a lack of sufficient evidence and application guidelines for this dietary approach. One of the challenges lies in the variation of food availability even within the same country, which affects the consistency of the diet.
Additionally, there have not been enough studies to establish clear, robust conclusions regarding its effectiveness in combating MASLD.
Given these gaps, we are currently gathering the available evidence in an effort to encourage the generation of more research on this topic. This will be crucial for developing a more reliable understanding of the milpa diet potential and its impact on MASLD.
Reviewer 3 Report
Comments and Suggestions for Authors
The aim of the submitted review article is benefits of Milpa Diet for MASLD in Mesoamerican population. Having in mind, that MASLD is a public health issue, being not only a major cause of liver-related morbidity and mortality, but also an independent risk factor for the development of noncommunicable diseases, the topic of this article can be important and useful, especially regarding the role of lifestyle interventions in the treatment of this disorder. Two recently published review papers „Effects of Foods of Mesoamerican Origin in Adipose Tissue and Liver-Related Metabolism” and “Nutritional, bioactive components and health properties of the milpa triad system seeds (corn, common bean and pumpkin)” are focused on the bioactive components of Milpa diet and their possible potential in the prevention of liver diseases. Authors of this review paper tried to do something similar and highlight the benefits of this diet in MASLD. However, the main issue is that the benefits of bioactive components are not extensively discussed and separated into results obtained in preclinical and clinical studies. In addition, the title, the aim of the paper and the conclusion sound too ambitious and speculative having in mind that there is no randomized clinical trial or original clinical study that demonstrates the efficacy of the milpa diet on MASLD (previously NAFLD). Based on the fact that this is just a narrative review paper, the manuscript should be better organized, the quality has to be improved and the results of in vivo, in vitro and clinical studies regarding the bioactive components and fatty liver disease should be more clearly presented and discussed. I deeply encourage authors to change the title and reorganize the paper into introduction, current treatment of MASLD (which include lifestyle intervention and pharmacological and non-pharmacological approach), Milpa diet, bioactive components of Milpa diet, results of pre-clinical and clinical studies, benefits and limitation of current studies, conclusion and future perspectives. After careful review and the significant improvement of the paper I strongly believe that it would attract the attention in this filed.
Some other issues need to be addressed:
-The second paragraph in the introduction is copy paste from https://pmc.ncbi.nlm.nih.gov/articles/PMC10583766/
-The rest of the text (line 41-59) in the introduction is copy paste from https://journals.lww.com/cld/fulltext/2022/02000/epidemiological_and_genetic_aspects_of_nafld_and.8.aspx
-the text given in lines 127-132 has to be rewritten because it is copy paste from doi: 10.3390/nu16050574
-the sentence given in lines 277-282 has to be rewritten (copy paste from https://link.springer.com/article/10.1007/s13679-024-00597-6)
-Discussion section should be deleted and focused on benefits and disadvantages of milpa diet in contrast to other diet approaches (i.e. Mediterranean diet). It is well known that Mediterranean diet is gold standard until now.
Author Response
Comments 1: The aim of the submitted review article is benefits of Milpa Diet for MASLD in Mesoamerican population. Having in mind, that MASLD is a public health issue, being not only a major cause of liver-related morbidity and mortality, but also an independent risk factor for the development of noncommunicable diseases, the topic of this article can be important and useful, especially regarding the role of lifestyle interventions in the treatment of this disorder. Two recently published review papers „Effects of Foods of Mesoamerican Origin in Adipose Tissue and Liver-Related Metabolism” and “Nutritional, bioactive components and health properties of the milpa triad system seeds (corn, common bean and pumpkin)” are focused on the bioactive components of Milpa diet and their possible potential in the prevention of liver diseases. Authors of this review paper tried to do something similar and highlight the benefits of this diet in MASLD. However, the main issue is that the benefits of bioactive components are not extensively discussed and separated into results obtained in preclinical and clinical studies. In addition, the title, the aim of the paper and the conclusion sound too ambitious and speculative having in mind that there is no randomized clinical trial or original clinical study that demonstrates the efficacy of the milpa diet on MASLD (previously NAFLD). Based on the fact that this is just a narrative review paper, the manuscript should be better organized, the quality has to be improved and the results of in vivo, in vitro and clinical studies regarding the bioactive components and fatty liver disease should be more clearly presented and discussed. I deeply encourage authors to change the title and reorganize the paper into introduction, current treatment of MASLD (which include lifestyle intervention and pharmacological and non-pharmacological approach), Milpa diet, bioactive components of Milpa diet, results of pre-clinical and clinical studies, benefits and limitation of current studies, conclusion and future perspectives. After careful review and the significant improvement of the paper I strongly believe that it would attract the attention in this filed.
Response 1: The change of several titles was taken into consideration and we went deeper into the medical treatment, however, we consider that it is important to address the different nutrients with their respective evidence to make the article more fluent.
Comments 2:
Some other issues need to be addressed:
-The second paragraph in the introduction is copy paste from https://pmc.ncbi.nlm.nih.gov/articles/PMC10583766/
-The rest of the text (line 41-59) in the introduction is copy paste from https://journals.lww.com/cld/fulltext/2022/02000/epidemiological_and_genetic_aspects_of_nafld_and.8.aspx
Response 2: we modified those parts, thank you for your comment.
Comments 3:
-the text given in lines 127-132 has to be rewritten because it is copy paste from doi: 10.3390/nu16050574
Response 3: It was deleted as we talk about this same topics further on the article
Comments 4:
-the sentence given in lines 277-282 has to be rewritten (copy paste from https://link.springer.com/article/10.1007/s13679-024-00597-6)
Response 4: we edited the information
Comments 5: -Discussion section should be deleted and focused on benefits and disadvantages of milpa diet in contrast to other diet approaches (i.e. Mediterranean diet). It is well known that Mediterranean diet is gold standard until now.
Response 5: We agree with this comment, so we have reorganized the information and discussed this comparison.
Round 2
Reviewer 2 Report
Comments and Suggestions for Authors
The review concludes that adopting the milpa diet could be a culturally relevant, nutritious, and sustainable dietary approach to preventing and treating MASLD, promoting better liver health and reducing the risk of associated diseases.
The paper has been improved in organization and theory depth to support the ideas or conclusions. Some questions need to be further considered before accepted.
- In the abstract, coffee consumption and alcohol is underscored, but they are not tightly related to the title.
- In the introduction part, there are too much paragraphs, some of them can be integrated.
- In the section of Current treatment for MASLD, probiotics and gut microbiota regulation should be supplemented as one of important treatment of MASLD through regulating bile acid metabolism, intestinal barrier, inflammation. A literature is recommended to be cited and support the lipid metabolism regulation based on bile acid alteration in high-fat intake: Cai H, Zhang J, Liu C, et al. High-fat diet-induced decreased circulating bile acids contribute to obesity associated with gut microbiota in mice[J]. Foods,2024,13(5).
- In figure 1 and 2, are those food fixed or free combinations? And what is the relationship or difference of these 2 figures with table 2?
- Figure 2 has too much blank.
- In the section of 4, gut microbiota regulation should be discussed, especially the benefit of carbohydrate in regulating gut microbiota.
Author Response
Comments 1: In the abstract, coffee consumption and alcohol is underscored, but they are not tightly related to the title.
Response 1: This part was removed. Thanks.
Comments 2: In the introduction part, there are too much paragraphs, some of them can be integrated.
Response 2: We integrated some paragraphs.
Comments 3: n the section of Current treatment for MASLD, probiotics and gut microbiota regulation should be supplemented as one of important treatment of MASLD through regulating bile acid metabolism, intestinal barrier, inflammation. A literature is recommended to be cited and support the lipid metabolism regulation based on bile acid alteration in high-fat intake: Cai H, Zhang J, Liu C, et al. High-fat diet-induced decreased circulating bile acids contribute to obesity associated with gut microbiota in mice[J]. Foods,2024,13(5).
Response 3: Thanks for the suggestion, we have added the study to our paper.
Comments 4: In figure 1 and 2, are those food fixed or free combinations? And what is the relationship or difference of these 2 figures with table 2?
Response 4: These examples of foods are free combinations, keeping the food groups on the plate according to the variety available; we do not refer to fixed combinations. Table 2 is a list of examples of foods containing fiber in the milpa diet; some of them are used in the images.
Comments 5: Figure 2 has too much blank
Response 5: The figure was modified
Comments 6: In the section of 4, gut microbiota regulation should be discussed, especially the benefit of carbohydrate in regulating gut microbiota.
Response 6: In the article, in the part where amaranth is mentioned, the role it has on the microbiota is discussed.
Reviewer 3 Report
Comments and Suggestions for Authors
General comments:
Although authors made some improvements, especially regarding the discussion part, which is mainly now focused on limitation of application of Mediterranean diet in contrast to Milpa diet the main issue is still present: The benefits of bioactive components are not extensively discussed and separated into results obtained in preclinical and clinical studies. I encourage authors to insert at least one table that summarizes the results obtained in the available studies. In addition, the title, the aim of the paper and the conclusion are still too ambitious and speculative having in mind that there is no randomized clinical trial or original clinical study that demonstrates the efficacy of the milpa diet on MASLD (previously NAFLD). Please rewrite the aim of the study because you did not discuss the pathophysiological benefits of the diet, you discuss benefits of its bioactive components.
Minor issues:
-In the section Current treatment for MASLD authors should emphasize that there is no definite treatment for MASLD.
-The sentence “For example, a study realized by Domínguez-Uscanga et al.[22] reported the beneficial effect of a baked snack containing 70% corn and 30% beans in reducing serum lipids by inhibiting PPAR-γ and SREBP2 in a murine model with a high-fat diet. The study suggests a reduction in obesity, dyslipidaemia, and MASLD.” Should be avoided in this part and this reference should be cited in the part where you discuss diet components.
Author Response
Comments 1: Although authors made some improvements, especially regarding the discussion part, which is mainly now focused on limitation of application of Mediterranean diet in contrast to Milpa diet the main issue is still present: The benefits of bioactive components are not extensively discussed and separated into results obtained in preclinical and clinical studies. I encourage authors to insert at least one table that summarizes the results obtained in the available studies. In addition, the title, the aim of the paper and the conclusion are still too ambitious and speculative having in mind that there is no randomized clinical trial or original clinical study that demonstrates the efficacy of the milpa diet on MASLD (previously NAFLD). Please rewrite the aim of the study because you did not discuss the pathophysiological benefits of the diet, you discuss benefits of its bioactive components.
Response 1:
The title of this paper is not intended as a recommendation or a conclusion; the authors do not consider it to be overly ambitious.What we do consider is that there are potential benefits of the components of the Milpa diet and it is more feasible to apply it in our population than mediterranean diet. Our review focuses on these potential advantages of milpa diet and we make some examples of factible applications as clinicians. We know there are no randomized clínicas trials and we discuss this point at the end of the paper under the title future perspectives. Given the lack of clinical studies and the above mentioned, this could interest other researchers to conduct clinical trials in this field.
According to your recommendation, we modified the aim of the study. In section 4 “Potential benefits of milpa diet components” we emphasize the nutritional characteristics of the milpa diet, its bioactive components and the benefits that its consumption may have in patients with MASLD.
Comments 2: In the section Current treatment for MASLD authors should emphasize that there is no definite treatment for MASLD.
Response 2: Thanks, it was added in the text.
Comments 3: The sentence “For example, a study realized by Domínguez-Uscanga et al.[22] reported the beneficial effect of a baked snack containing 70% corn and 30% beans in reducing serum lipids by inhibiting PPAR-γ and SREBP2 in a murine model with a high-fat diet. The study suggests a reduction in obesity, dyslipidaemia, and MASLD.” Should be avoided in this part and this reference should be cited in the part where you discuss diet components.
Response 3: Thank you for your recommendation, we moved the reference to the discuss section.
Round 3
Reviewer 3 Report
Comments and Suggestions for Authors
General comments:
Authors made some improvements and inserted the main findings obtained in pre-clinical and clinical studies regarding the constituents of Milpa diet in a table. However, they decided not to change title as well as conclusion. I still believe that conclusion is too speculative and not focused on the summarized results. The abstract is also too general.
Additional comments
-In the abstract, authors should rewrite the last sentence because this is just a narrative review paper not a study.
-Again, please rewrite the aim of the study. Academic writing usually uses an impersonal tone, so avoid „we“, „our“…
-The sentence given in lines66-67 should be rewritten. Please pay attention on English language as well as the fact that there is no definite pharmacological treatment for MASLD. So, lifestyle interventions, including diet and physical activity, are still the main approach.
-The sentence given in line 171 should be rewritten. For example: The main findings of preclinical and clinical studies focused on bioactive compounds found in Milpa Diet are summarized in Table S1.
-The Table S1 caption should be rewritten. For example: The effects of Milpa Diet compounds in MAFLD -main results of preclinical and clinical studies
-Please uniform Table S1 in the way that for each compound first authors should present results of pre-clinical and after that results of clinical study.
Author Response
Comments 1: Authors made some improvements and inserted the main findings obtained in pre-clinical and clinical studies regarding the constituents of Milpa diet in a table. However, they decided not to change title as well as conclusion. I still believe that conclusion is too speculative and not focused on the summarized results. The abstract is also too general.
Response 1: Thank you very much for your recommendations. The abstract and conclusions were modified accordingly.
Comments 2: -In the abstract, authors should rewrite the last sentence because this is just a narrative review paper not a study.
Response 2: In the abstract, the last sentence was modified.
Comments 3: -Again, please rewrite the aim of the study. Academic writing usually uses an impersonal tone, so avoid „we“, „our“…
Response 3: The aim of the study was rewritten using an impersonal tone.
Comments 4: -The sentence given in lines66-67 should be rewritten. Please pay attention on English language as well as the fact that there is no definite pharmacological treatment for MASLD. So, lifestyle interventions, including diet and physical activity, are still the main approach.
Response 4: The sentence given in lines 66-67 was rewritten.
Comments 5:-The sentence given in line 171 should be rewritten. For example: The main findings of preclinical and clinical studies focused on bioactive compounds found in Milpa Diet are summarized in Table S1.
Response 5: The sentence given in line 171 was rewritten.
Comments 6: -The Table S1 caption should be rewritten. For example: The effects of Milpa Diet compounds in MAFLD -main results of preclinical and clinical studies
Response 6: The caption in table S1 was rewritten.
Comments 7: -Please uniform Table S1 in the way that for each compound first authors should present results of pre-clinical and after that results of clinical study
Response 7: The Table S1 was reorganized, presenting the preclinical studies first and after the clinical studies (when available).
Round 4
Reviewer 3 Report
Comments and Suggestions for Authors
The authors made significant improvements in the text and the quality of this narrative review paper is much better in comparison to the previous versions of this paper.
I hope that this narrative review will attract the attention of readers.